# Evaluation of a Virtual Reality implementation of a binocular imbalance test

**Santiago Martín**[1]*, **Juan A. Portela**[2], **Jian Ding**[3], **Oliver Ibarrondo**[4], **Dennis M. Levi**[3]

**1** University of Oviedo, Oviedo, Asturias, Spain, **2** Department of Optometry, Clinic Begira, Bilbao, Spain, **3** School of Optometry, University of California, Berkeley, CA, United States of America, **4** OSI Alto Deba, Unidad de Investigación AP, Arrasate, Spain

* martinsantiago@uniovi.es

**Data Availability Statement:** All relevant data are within the manuscript and its Supporting Information files.

**Funding:** SM received a mobility grant 2017 from the University of Oviedo (www.uniovi.es), and DML

## Abstract

The purpose of this study was (1) to implement a test for binocular imbalance in a Virtual Reality headset, (2) to assess its testability, reliability and outcomes in a population of clinical patients and (3) to evaluate the relationships of interocular acuity difference, stereoacuity and binocular imbalance to amblyogenic risk factors. 100 volunteers (6 to 70 years old, mean 21.2 ± 16.2), 21 with no amblyogenic risk factors and 79 with amblyopia or a history of amblyopia participated. Participants were classified by amblyogenic risk factor (24 anisometropic, 25 strabismic and 30 mixed) and, for those with strabismus, also by refractive response (16 accommodative and 39 non-accommodative). We characterized our sample using three variables, called the 'triplet' henceforth: interocular acuity difference, stereoacuity and imbalance factor. Binocular imbalance showed high test-retest reliability (no significant difference between test and retest in a subgroup, n = 20, p = 0.831); was correlated with Worth 4 dots test (r = 0.538, p<0.0001); and correlated with both interocular acuity difference (r = 0.575, p<0.0001) and stereoacuity (r = 0.675, p<0.0001). The mean values of each variable of the triplet differed depending on group classification. Mixed and non-accommodative groups showed the worst mean values compared with the other groups. Among participants with strabismus, strabismic vs mixed subgroups did not show significant differences in any variable of the triplet, whereas the accommodative vs non-accommodative subgroups showed significant differences in all of them. According to a univariate logistic model, any variable of the triplet provides a good metric for differentiating patients from controls, except for binocular imbalance for anisometropic subgroup. The proposed binocular imbalance test is feasible and reliable. We recommend monitoring amblyopia clinically not only considering visual acuity, but also stereoacuity and interocular imbalance. Stereoacuity on its own fails because of the high percentage of patients with no measurable stereoacuity. Binocular imbalance may help to fill that gap.

## Introduction

Amblyopia is a neuro-developmental disorder of the visual cortex that arises from abnormal visual experience early in life and leads to reduced visual acuity (generally in one eye) [1].

received a grant from the National Eye Institute (https://www.nei.nih.gov/) R01 EY020976. The funders had no role in study design, data collection and analysis, decision to publish, or preparation of the manuscript.

**Competing interests:** I have read the journal's policy and the authors of this manuscript have the following competing interests: SM and JP promoted, with the support of the University of Oviedo, the creation of the startup VisionaryTool. Both have assisted VisionaryTool, S.L. (www.visionarytool.com) to create a commercial version of the VR imbalance test described in this manuscript (University of Oviedo contract FUO-EM-19-099). VisionaryTool has not had any role (writing, analysis, or control over publication) in the production of the paper. This does not alter our adherence to PLOS ONE policies on sharing data and materials.

Amblyopia is clinically relevant because it affects between 1% and 4% of the general population [2]. Currently, treatment for amblyopia consists primarily of refractive error correction and penalization or occlusion of the strong eye [3].

Despite the fact that the diagnosis and treatment of amblyopia are defined in terms of visual acuity, a host of other visual functions are affected, both monocular and binocular [3,4]. Amblyopia is associated with strabismus, anisometropia or their combination [5]. These clinical categories manifest differences in the pattern of visual loss [6,7]. Amblyopia arises from the mismatch between the images captured by each eye; the information from one eye is favored, while input from the other eye is suppressed [8]. Both suppression and reduced stereo acuity are common characteristics of amblyopia [1,6,9–12]. Some (mainly strabismic) amblyopes, fail standard clinical stereotests (i.e., they cannot respond to the largest disparity). The absence of an initial measure of stereoacuity makes quantification difficult [1,13]. Additionally, the variability of stereoacuity measurements (as much as a factor of 4) [14] adds uncertainty. Moreover, different tests often give different thresholds, increasing practitioner confusion [15,16]. Successful treatment of amblyopia requires improving both the visual acuity of the amblyopic eye, and binocular vision. However, monitoring just acuity and stereoacuity might not be sufficient given the issues described above. Recent findings suggest that suppression rather than visual acuity loss limits stereoacuity in observers with amblyopia, and stereopsis improves when interocular dominance is neutralized [17]. Thus, the concept of suppression has been proposed as a third leg to monitor amblyopia treatment [18,19].

Clinically, the presence and extent of the suppression scotoma is commonly assessed using the Bagolini striated lens test [19] or the Worth 4 dot test [20]. Nevertheless, recent research has pointed out the importance of measuring and quantifying the severity of suppression or interocular imbalance and several specific tests have been proposed based on determining the ratio of contrast or luminance in the two eyes that "balances" the binocular input to the primary visual cortex [21–25].

There have been several different laboratory approaches to measuring the binocular imbalance in patients with amblyopia [9,11,21–27]. Many of these tests require special equipment to present different images to the two eyes [11,19]. However, variations have been developed for use with Tablets and colored filters [28,29].

Kwon et al [27] proposed a test to assess binocular imbalance as a function of spatial frequency using a dichoptic letter chart. Firstly, they create a set of spatial frequency band-pass filtered Sloan letters. At each position of the dichoptic chart, the identity and interocular contrast-ratio of the letter differs while the spatial-frequency content of the letter remains the same. Participants, wearing stereo-shutter glasses, read the dichoptic letters out loud, and the balance point is calculated as the interocular contrast-ratio that gives equal probability of reporting the letters perceived in the two eyes. We found this study particularly interesting, firstly because it considers the relationship between spatial frequency and binocular imbalance degree, as found in previous studies [23] and secondly because it has proven its clinical feasibility [21].

One major drawback of tablet tests is that they may not allow full compensation for misalignment of the images in the two eyes that occur in patients with strabismus. Ding & Levi [30] point out that achieving binocular alignment and fusion might be the first step in the recovery of stereopsis, and for this purpose they design a dichoptic cross with binocular fusion locks, a surrounding high-contrast frame, and four luminance squares, viewed through a custom four-mirror stereoscope. One of the strabismic participants in their study achieved stereopsis through the binocular combination task alone, with no stereo training.

Image alignment can be accomplished via software in Virtual Reality (VR). Indeed, Black et al. [22] implemented an adjustment for vertical and horizontal misalignments and a

dichoptic binocular imbalance test in a VR headset using a dichoptic cross with binocular fusion locks. Importantly, they used the results (alignment and binocular balance settings) as a starting point for a training program. VR has the potential to replace the Synoptophore (or Major Amblyoscope), as the standard instrument for the assessment and treatment of ocular motility disorders. VR has been used successfully in the treatment of mechanical strabismus and amblyopia [22,31–34]. The recent incorporation of eye-tracking technology to VR can potentially allow the measurement of deviation angles objectively, as a preliminary study suggests [31].

As far as we know, the feasibility of using VR as part of routine clinical assessment of amblyopia has not been investigated yet. This motivates the present study, which we conceive to be a first step towards the design of new VR based method of assessment of binocular balance in amblyopia which can be used as a starting point for treatment.

The aim of this study was three-fold: 1) To implement a test for binocular imbalance in a Virtual Reality headset, 2) To assess the testability, reliability and outcomes of this test in a population of clinical patients and 3) To evaluate the relationship of interocular acuity difference, stereoacuity and binocular imbalance to amblyogenic risk factors (strabismus, anisometropia or their combination, referred to as 'mixed') in a large population of clinical patients.

## Material and methods

### Participants

All volunteers were recruited at the same Optometry Clinic. The data was obtained in their first visit to the clinic, after signing the Consent Agreement (minors signed the agreement together with their parents). The protocol was approved by the Regional Ethics Committee of Clinic Research (Asturias, Spain) and follows the Helsinki Declaration.

We enrolled 100 volunteers (ages 6 to 70 years old, mean 21.2 ± 16.2 years) (S1 Table). Of those patients, 21 volunteers had no amblyogenic factor (control group). Seventy nine have amblyopia or a history of amblyopia (24 anisometropic, 25 strabismic and 30 mixed), most of whom had received previous treatments: all of them refractive correction; 59 occlusion; 35 perceptual learning using Gabor patches to improve contrast sensitivity; 24 perceptual learning using random dot stimuli to improve stereoacuity [35]; and six subjects strabismus surgery.

All 55 participants with strabismus had esotropia. Exotropia is less prevalent than esotropia [36] and is much more likely to be intermittent exotropia [37]. In intermittent exotropia, binocular inhibition is low [38], and the deviation angle at near is generally lower than at far distances [39]. As a result, stereoacuity is likely to be preserved, and amblyopia is uncommon. This may explain why in our strabismus sample we find only participants with esotropia. Participants were also classified according to their response to optical correction: 16 were classified as accommodative (the deviations disappears at near and far distance with full optical correction condition) and 39 as non-accommodative (residual esotropia at near and/or far distance despite wearing full optical correction) [40].

Amblyopia was defined as ≥0.10 logMAR best-corrected visual acuity in the amblyopic eye and interocular difference of ≥0.2 logMAR. Anisometropia was defined as an amblyogenic factor due to a spherical equivalent interocular difference of ≥ 1.0 D [41]. The spherical equivalent was calculated as the sum of sphere plus half the cylinder. Strabismus was defined as an amblyogenic risk factor based on the presence of heterotropia at near or far distance, measured with the Unilateral Cover Test (UCT), with full optical correction condition and an accommodative stimulus [42]. Mixed amblyogenic risk factor was defined as the presence of both strabismus and anisometropia.

Exclusion criteria were congenital malformation, ocular pathology, concurrent treatment with atropine penalization, presence of diplopia in daily life conditions, prematurity ≥8 weeks,

developmental delay, and coexisting ocular or systemic disease. Due to Virtual Reality headset limitations, volunteers with an interpupillary distance less than 55 mm and/or lower head circumference less than 500 mm were also excluded [43,44].

## Clinical protocol

The same optometrist (author J.A.P.) evaluated all participants. Visual evaluation included: Best Corrected distance Visual Acuity (BCVA) on a logarithmic visual acuity chart (LogMAR acuity) with a polarized screen (SmarThing4vision, Spain); UCT using accommodative stimuli; refractive error by autorefractor under cycloplegia (cyclopentolate 1%) (Topcon model TRK 1P); and slitlamp and ophthalmoscopic evaluation of the anterior and posterior segment.

Binocular vision was evaluated in three different ways. Firstly, we registered patient responses (fusion, suppression or diplopia) using the Worth 4 dot test with a polarized screen, at a distance of 4 meters and for two target sizes (visual angles of 1.5˚ and 5.0˚), following the test manufacturer's instructions (SmarThing4Vision, Spain). The Worth 4 dot test was carried out without prisms in strabismic patients to facilitate the detection of diplopia.

Secondly, binocular imbalance was measured using the VR test implemented in this paper and described below. To evaluate test-retest reliability, 20 volunteers repeated the test in a second routine visit to the clinic. Inclusion criteria for the second test were (1) no visual therapy activities between the two visits and (2) no change in visual acuity or stereoacuity. Selected volunteers represented all visual conditions.

Thirdly, we measured stereoacuity using the Randot Preschool Stereoacuity Test (RPST) conducted according to the test manufacturer's instructions (Stereo Optical, USA). Measurements were transformed from seconds of arc to log10 units for the study. A value of 3.11 in log10 units (1300 arc seconds, which Chopin et al defined as "ecological stereoblindness", was assigned to patients without measurable stereoacuity [45].

## Binocular imbalance test in VR

Binocular imbalance was assessed using a modified version of the dichoptic eye chart proposed by Kwon et al [27] implemented in a Virtual Reality (VR) device (Vive by HTC Co.).

The test uses letters taken from the Sloan font alphabet and filtered to 3 cycles per letter size (cosine log filter) [46,47]. Letters are normalized to device mean luminance and root-mean-square (RMS) contrast of 0.1. The size of the letter determines the frequency to be tested in the VR headset. Due to the resolution restrictions of the headset, the frequency selected is 0.68 cpd. Letters are located at a distance of 2.0 m in virtual space (i.e. minimal accommodation and convergence).

Importantly for strabismic patients who are unable to fuse the images, prior to the imbalance test a dichoptic nonius alignment screen is presented (Fig 1, S1 Movie). It consists of a high contrast square crossed by two lines, horizontal and vertical. Each line has four segments, two inside the square, and two outside (one on each side). Each segment is seen only by one eye. Any perceived misalignment (horizontal, vertical or cyclo) is adjusted using the software, until the patient reports the correct cyclopean perception.

The test starts with a dichoptic eight squares pattern inside a high contrast frame subtending 20˚, which remains visible during the test to facilitate fusion. The participant has to confirm seeing the pattern stable before starting each trial. On each trial, two randomly selected dichoptic letters are presented to the observer (one to each eye) for 200 msec., and the participant is instructed to report the letter seen (or the dominant letter in case of both are perceived simultaneously), following which both previous letters are presented side-by-side, enhanced in contrast and size, and the observer must reaffirm the one that was seen. This response is

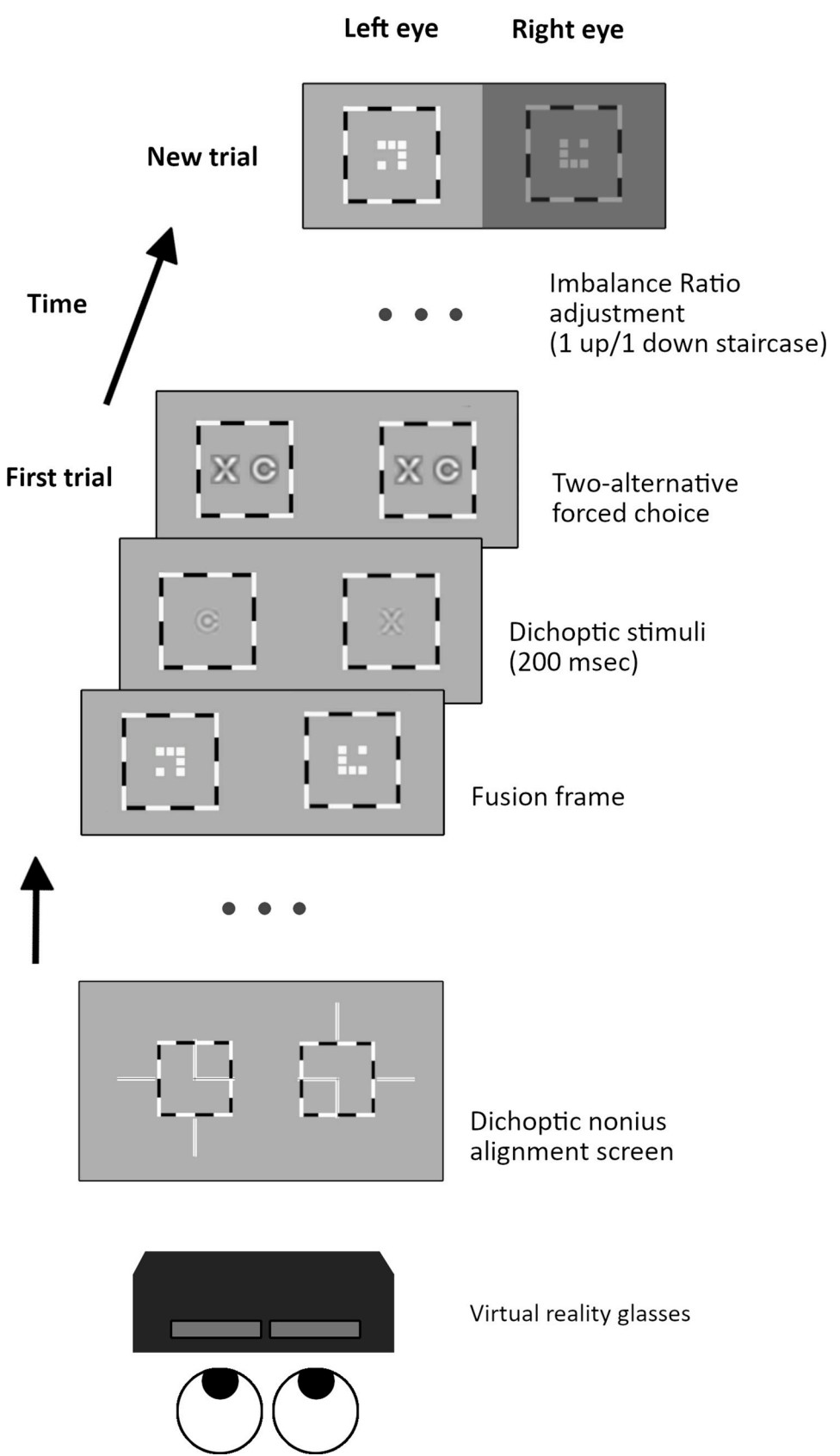

**Fig 1. Binocular imbalance test.** Prior to the test itself, a dichoptic nonius alignment screen is presented (bottom). The test consists of a series of trials, each one divided into three steps (from bottom to top): firstly, a fusion frame of a dichoptic nine square grid surrounded by a high contrast frame is presented to facilitate fixation. After the experimenter presses the spacebar, the computer displays two dichoptic letters, randomly selected and filtered to the target interocular contrast, for 200 msec. Finally, both letters are presented binocularly one next to the other with enhanced contrast and size. The observer's task is to indicate which letter he/she has perceived (two-alternative forced choice). The computer adjusts the imbalance ratio following a 1 up/ 1 down staircase for the next trial, until a valid threshold is obtained.

recorded by the experimenter. Stimulus duration (200msec) is set to avoid possible eye-movement effects on interocular alignment in people with strabismus.

The test follows a 1 up/1 down staircase: on each new trial, image contrast is increased for the eye that does not see the letter and decreased by the same amount for the eye that has just seen the letter. After a maximum of 16 reversals or 40 trials, the test is complete (lasting around 2 minutes). If there have been more than nine reversals, the test is considered valid. We define the mean contrast value of the last four reversals as the Imbalance Ratio (IR), i.e. fellow eye contrast divided by dominant eye contrast at which there is equal probability of reporting the optotype presented to any eye. An imbalance ratio of 1.0 means that binocular vision is perfectly balanced, whereas a ratio of 3.0 means that the amblyopic eye needs 3.0 times more contrast than the fellow eye to achieve balanced vision.

Patients can perform the test while seated, reducing the risk of VR dizziness of the fatigue due to helmet weight (approx. 500 grams). The interpupillary distance adjustment range of the headset is limited to approx. 61 to 73mm and constitutes an important limitation.

## Statistical analysis

We used R-Statistic (v.3.6.0) to calculate descriptive statistics. Wilcoxon signed rank test with continuity correction was used to evaluate test-retest reliability. Statistical differences between the means were calculated using the t-student test for variables that follow a normal distribution and the Mann-Whitney test is used for variables that do not conform to a normal distribution. The joint comparison of multiple groups was made using ANOVA in the case of normally distributed variables and the Kruskal-Wallis test when the variables were not normally distributed. The analysis allows us to evaluate differences between the groups (amblyogenic factor and strabismus type). Comparison of correlations between different variables was calculated using the Spearman test.

Finally, we used a logistic model to calculate the probability of belonging to different groups and to establish a 0.5 cut off point. The analysis conducts a univariate analysis to study the capacity of each classification variable in the triplet (acuity difference, imbalance factor and stereoacuity) to differentiate the groups defined by amblyogenic factor (control, anisometropia, strabismus and mixed) and refractive response (accommodative and non-accommodative).

## Results

All 100 volunteers, including 18 between the ages of 6 and 8, were able to complete the binocular imbalance test. No one reported dizziness or fatigue due to helmet weight. All participants appear to have understood the test procedure after a short explanation.

The binocular imbalance test is highly reliable. Test-retest repeatability (Wilcoxon signed rank test), gave a p-value of 0.831, showing no significant difference between first and second test. A Bland-Altman plot is included in Fig 2. The mean of differences is equal to -0.033, which means that first measurements are 1.033 times bigger than second results. Differences follow a normal distribution (Shapiro test, p = 0.001).

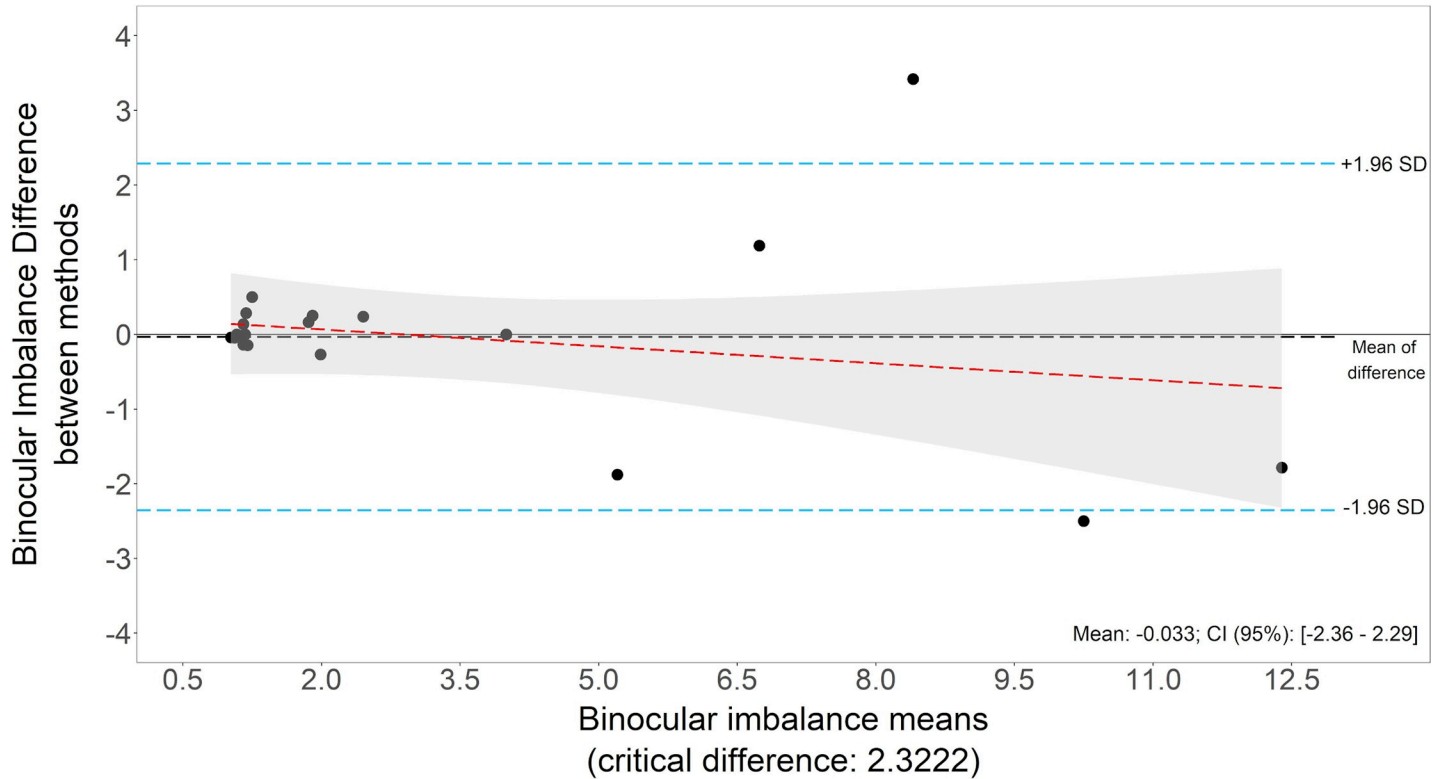

**Fig 2. Binocular imbalance repeatability Bland-Altman plot.** Binocular imbalance results in first and second measurements. N = 20 volunteers.

To assess whether the binocular imbalance test results were consistent with standard clinical measures of suppression, we evaluated its correlation with the Worth 4 dots test. The Worth test provides patient's responses (fusion, suppression or diplopia) at two visual angles (1.5˚ and 5.0˚). These responses were ordered according to the extent of suppression scotoma, considering diplopia as an intermediate stage between fusion and suppression, obtaining a novel scale of six categories (Table 1). We conjecture that, if diplopia occurs with the Worth test and not in daily life conditions, it seems likely that there is suppression under normal

**Table 1. Correlation between Worth 4 dot test and triplet variables.** Mean and Standard Deviation for each variable of the triplet (interocular visual acuity difference, stereoacuity and imbalance factor) for each category considered of Worth test. Worth test answers have been ordered according to the extent of suppression scotoma, considering diplopia as an intermediate stage between fusion and suppression. Number of occurrences in the sample, N, is also included. Correlations and p-values between Worth test categories and each variable of the triplet are included in the last row.

| Worth test categories | N | Acuity difference | | Imbalance Factor | | Stereoacuity | |
|---|---|---|---|---|---|---|---|
| | | mean | SD | mean | SD | mean | SD |
| 0 (1.5 F + 5.0 F) | 57 | 0.04 | 0.07 | 1.54 | 0.75 | 1.91 | 0.40 |
| 1 (1.5 D + 5.0 F) | 1 | 0.00 | | 1.63 | | 2.90 | |
| 2 (1.5 D + 5.0 D) | 10 | 0.03 | 0.04 | 2.65 | 1.87 | 3.11 | 0.00 |
| 3 (1.5 S + 5.0 F) | 10 | 0.21 | 0.13 | 1.88 | 0.88 | 2.75 | 0.46 |
| 4 (1.5 S + 5.0 D) | 15 | 0.37 | 0.24 | 6.29 | 4.40 | 3.10 | 0.05 |
| 5 (1.5 S + 5.0 S) | 7 | 0.72 | 0.25 | 7.64 | 5.02 | 3.04 | 0.19 |
| Correlation (p-value) | | 0.696 (p < 0.0001) | | 0.538 (p < 0.0001) | | 0.791 (p < 0.0001) | |

F: fusion; D: diplopia; S: suppression; 1.5 degrees and 5.0 degrees of visual angle.

**Table 2. Correlation between triplet variables at each clinic group.** Correlation values (Spearman test) between pairs of the continuous variables of the study (acuity difference, stereoacuity and imbalance factor), considering all patients and each group individually, classified according to amblyogenic factor (anisometropia, strabismus and mixed) and considering refractive response (accommodative and non-accommodative).

| Comparison | All patients | Anisometropia | Strabismus | Mixed | Accommodative | Non-accomm. |
|---|---|---|---|---|---|---|
| Acuity difference vs Imbalance factor | 0.575 (p < 0.01)* | 0.390 (p = 0.06) | 0.532 (p = 0.01)* | 0.666 (p < 0.01)* | 0.361 (p = 0.17) | 0.664 (p < 0.01)* |
| Acuity difference vs Stereoacuity | 0.601 (p < 0.01)* | 0.678 (p < 0.01)* | 0.428 (p = 0.03)* | 0.412 (p = 0.02)* | 0.168 (p = 0.53) | 0.251 (p = 0.12) |
| Imbalance factor vs Stereoacuity | 0.675 (p < 0.01)* | 0.421 (p = 0.04)* | 0.395 (p = 0.05) | 0.560 (p < 0.01)* | 0.154 (p = 0.57) | 0.332 (p = 0.04)* |

* Significant correlation because p-value is less than 0.05.

everyday conditions, but the suppression scotoma is weak or small. For example, a subject could have diplopia at 5.0° and suppression at 1.5°. Alternatively, the patient might fuse at 5.0° degrees and suppress at 1.5°, but never fuse or exhibit diplopia at 1.5° and suppress at 5° degrees. Diplopia appears due to the highly dissociative stimuli used in Worth test. The six levels of suppression are obtained this way were highly correlated with the result of the binocular imbalance test (r = 0.538, p<0.0001). There was also a significant correlation between Worth test and stereoacuity (r = 0.791, p<0.0001), and between Worth test and interocular visual acuity difference (r = 0.696, p<0.0001).

Moreover, as summarized in Table 2, the binocular imbalance test correlated with LogMAR interocular visual acuity difference (r = 0.575, p<0.0001) and stereoacuity (r = 0.675, p<0.0001). LogMAR acuity difference and stereoacuity were also correlated (r = 0.601, p<0.0001). Those correlations were positive when analyzing the whole dataset, and also occur in some cases when considering the clinic subgroups of patients, by amblyogenic risk factor and by refractive response.

We assessed the sensitivity of the binocular imbalance test for detecting inter-observer differences based on amblyogenic risk factors: control, anisometropia, strabismus and mixed. Additionally, we classified the volunteers included in the strabismus and mixed groups according to their refractive response: accommodative and non-accommodative. Consistent with previous studies [19,21], the mixed group clearly shows the worst visual acuity difference, stereoacuity and imbalance factor (Table 3). Among patients in the strabismus and mixed groups, the non-accommodative group performing poorly on all 3 measures. This can be seen clearly in Fig 3, which shows the distribution of the triplet variables using box plots for each group. Importantly, only 1 accommodative patient out of 16 is stereoblind, whereas 33 out of 39 non-accommodative patients are stereoblind.

**Table 3. Mean and Standard Deviation for triplet variables.** Interocular visual acuity difference, stereoacuity and imbalance factor mean and standard deviation for each group, classified according to amblyogenic factor (control, refractive, strabismus and mixed) and considering refractive response (accommodative and non-accommodative).

| | | Acuity difference | | Imbalance factor | | Stereoacuity | |
|---|---|---|---|---|---|---|---|
| | N | Mean | SD | Mean | SD | Mean | SD |
| *Amblyogenic factor* | | | | | | | |
| Control | 21 | 0.000 | 0.000 | 1.246 | 0.152 | 1.648 | 0.123 |
| Anisometropia | 24 | 0.129 | 0.165 | 1.506 | 0.660 | 2.085 | 0.501 |
| Mixed | 25 | 0.246 | 0.270 | 4.395 | 3.925 | 2.724 | 0.511 |
| Strabismic | 30 | 0.191 | 0.288 | 3.540 | 3.542 | 2.865 | 0.474 |
| *Refractive response* | | | | | | | |
| Accommodative | 16 | 0.073 | 0.072 | 1.861 | 1.066 | 2.092 | 0.356 |
| Non-accommodative | 39 | 0.282 | 0.307 | 4.887 | 4.094 | 3.074 | 0.105 |

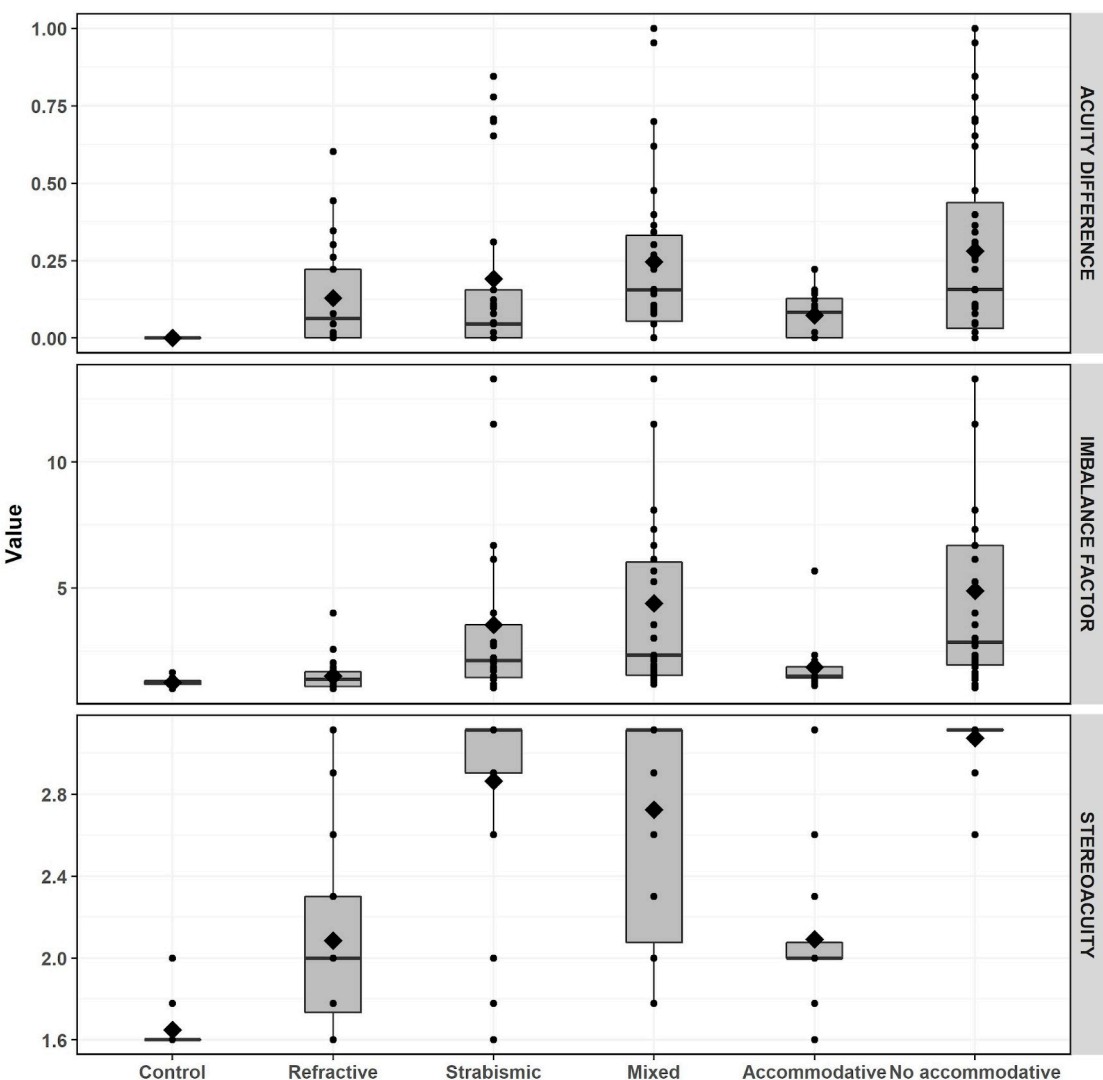

**Fig 3. Triplet variables box plots.** Graphical display of the continuous variables (LogMAR acuity difference, log10 stereoacuity and imbalance factor) using boxplots for each classification groups: amblyogenic factor (control, anisometropic, strabismus and mixed) and deviation nature (accommodative and non-accommodative). The box plot shows the mean and distribution in quartiles of the data. Circles represents each single participant.

To delve more deeply into the differences between groups, we carried out a means paired comparison (Table 4). Both group classifications, amblyogenic factor and refractive response, were analyzed. Whereas the strabismus and mixed groups show no significant mean differences in any of the variables, accommodative and non-accommodative groups show significant differences in all three variables. Binocular imbalance is the only variable that shows significant differences between the accommodative and anisometropia groups (p = 0.042). On the other hand, imbalance is the only variable of the triplet that fails to differentiate control vs anisometropia groups. Finally, interocular acuity difference is the only variable of the triplet that fails to differentiate the anisometropia group from the mixed (p = 0.093) and non-accommodative groups (p = 0.063).

The univariate logistic models allow a comparison of the significance of each continuous variable to differentiate any group considered of patients from controls (Table 5). The

**Table 4. Means pair comparison between group pairs for each triplet variable.** P-value and significance (p-value< 0.05) for each triplet variable (visual acuity difference, stereoacuity and imbalance factor) per each group classification pair. Both group classifications are considered, amblyogenic factor (control, refractive, strabismus and mixed) and refractive response (accommodative and non-accommodative).

| Groups | Acuity diffrence | | Stereoacuity | | Imbalance factor | |
|---|---|---|---|---|---|---|
| | p.Value | Sig | p.Value | Sig | p.Value | Sig |
| Control vs Anisometropia | < 0.01 | * | < 0.01 | * | 0.33 | |
| Control vs Strabismic | < 0.01 | * | < 0.01 | * | < 0.01 | * |
| Control vs Mixed | < 0.01 | * | < 0.01 | * | < 0.01 | * |
| Mixed vs Anisometropia | 0.09 | | < 0.01 | * | < 0.01 | * |
| Mixed vs Strabismic | 0.15 | | 0.37 | | 0.26 | |
| Anisometropia vs Strabismic | 0.81 | | < 0.01 | * | 0.001 | * |
| Accommodative vs Control | < 0.01 | * | < 0.01 | * | < 0.01 | * |
| Control vs Non-accommodative | < 0.01 | * | < 0.01 | * | < 0.01 | * |
| Accommodative vs Non-accomm | 0.03 | * | < 0.01 | * | < 0.01 | * |
| Accommodative vs Anisometropia | 0.54 | | 0.45 | | 0.04 | * |
| Non-accomm vs Anisometropia | 0.06 | | < 0.01 | * | < 0.01 | * |

* Significant correlation because p-value is less than 0.05.

stereoacuity variable shows the ROC Area Under the Curve best results, above 0.9 in most cases, when compared to the rest of the univariate logistic models i.e. it provides the best differentiation of any group from controls (except anisometropia, with the lower ROC AUC value, 0.82, still high). The imbalance factor also provides strong differentiation, above 0.8 in most

**Table 5. Univariate logistic classification models.** Results obtained by three different models, based on acuity difference, stereoacuity and imbalance factor, when differentiating control group from others: amblyogenic factor (anisometropia, strabismus or mixed) and refractive response (accommodative or non-accommodative).

| | ROC AUC | Sensitivity | Specificity | PPV | NPV |
|---|---|---|---|---|---|
| **Control vs Anisometropia** | | | | | |
| Acuity difference | 0.83 | 1.00 | 0.67 | 0.72 | 1.00 |
| Stereoacuity | 0.82 | 0.86 | 0.75 | 0.75 | 0.86 |
| Imbalance factor | 0.59 | 0.52 | 0.54 | 0.50 | 0.57 |
| **Control vs Strabismic** | | | | | |
| Acuity difference | 0.84 | 1.00 | 0.68 | 0.72 | 1.00 |
| Stereoacuity | 0.97 | 0.90 | 0.92 | 0.90 | 0.92 |
| Imbalance factor | 0.83 | 0.95 | 0.72 | 0.74 | 0.95 |
| **Control vs Mixed** | | | | | |
| Acuity difference | 0.88 | 1.00 | 0.77 | 0.75 | 1.00 |
| Stereoacuity | 0.98 | 0.90 | 0.97 | 0.95 | 0.94 |
| Imbalance factor | 0.93 | 0.86 | 0.87 | 0.82 | 0.90 |
| **Control vs Accommodative** | | | | | |
| Acuity difference | 0.81 | 1.00 | 0.63 | 0.78 | 1.00 |
| Stereoacuity | 0.92 | 0.90 | 0.81 | 0.86 | 0.87 |
| Imbalance factor | 0.87 | 0.86 | 0.75 | 0.82 | 0.80 |
| **Control vs Non-accommodative** | | | | | |
| Acuity difference | 0.88 | 1.00 | 0.77 | 0.70 | 1.00 |
| Stereoacuity | 1.00 | 1.00 | 1.00 | 1.00 | 1.00 |
| Imbalance factor | 0.89 | 0.95 | 0.85 | 0.77 | 0.97 |

ROC.AUC: Area under the curve; PPV: Positive Predictive Value (PPV); NPV: Negative Predictive Value.

cases, better or similar to interocular acuity difference except for anisometropia group (ROC AUC value = 0.59).

## Discussion

The first aim of this study was to implement a test of binocular balance in Virtual Reality (VR) that could be used in a clinical setting. We conducted the test on 100 volunteers (including juveniles) with different visual conditions. The test procedure was easy to understand by patients, and test duration, around 2 minutes, was short enough to maintain attention even with younger volunteers. There were no reports of dizziness or fatigue due to helmet weight. Nevertheless, there are several limitations to consider. The first limitation is the limited range of interpupillary distance adjustment (approx. 61 to 73mm). Manufacturers should solve this limitation in future releases, as it affects an important percentage of the potential customers of VR products [43,48,49], particularly children younger than 7 years old. The wrong IPD adjustment can lead to incorrect judgment of depth and could potentially induce motion related sickness or disorientation.

Virtual Reality (VR) may become a promising tool for amblyopia treatment. This test could provide relevant information for setting the interocular contrast for perceptual learning or videogame play based in VR technology. However, due to a second limitation, the low resolution of VR headsets, binocular imbalance could only be tested at 0.68 cpd. Binocular imbalance due to amblyopia is more evident at high frequencies [24]. Thus, the discriminative power of the binocular imbalance test may be better if the test is conducted at higher frequencies, something that technology's natural evolution will provide us in the near future.

Although the purpose of the test is to measure binocular imbalance, the stimuli used would vary in size if the frequency tested were different. All tests based on dichoptic optotypes share this third limitation: the binocular imbalance is measured at a certain fixed frequency and size (visual angle) relationship. In our implementation, the optotype subtends a visual angle of 4.4˚. As the Worth 4 dot test results show, suppression is more evident at smaller visual angles. Thus, it would be desirable to perform the test at a smaller visual angle, i.e., at a higher frequency.

The overlapping optotypes approach used in the binocular imbalance test proposed has high test-retest reliability (intra-observer consistency) in previous implementations [50]. Here we confirmed the high test-retest reliability in our VR implementation.

Test validity was assessed using three complementary strategies. Firstly, we found a significant correlation between the binocular imbalance test and the Worth test results (Table 1). Although the Worth test measures the extent of the suppression scotoma rather than its intensity, it is reasonable to expect a correlation between the two. Our novel Worth test score assumes diplopia as an intermediate stage between fusion and suppression. Considering diplopia and performing the test at different sizes, we have a broader scale than the one proposed by Webber et al. [18]. They proposed a composite binocular function score derived from clinical stereoacuity measures and the Worth 4 Dot response at 33 cm; and found a high correlation with the inter-ocular contrast balance test proposed by Kwon et al. [27]. We have replicated that correlation, but between the Worth test alone (without combining it with stereoacuity data) and the binocular imbalance test. This avoids any bias in the result due to the already known correlation between stereoacuity and binocular imbalance. However, it should be noted that the binocular imbalance measure may not simply reflect suppression, since it does not separate out reduced monocular sensitivity from the effects of binocular suppression.

It also should be noted that it would have been desirable to perform the binocular imbalance test at a smaller visual angle and higher frequency. Our proposed Worth 4 Dot scale fails

when we compare the binocular imbalance results between categories 2 (mean 2.65 ±1.87) and 3 (mean 1.88 ±0.88). A subject with diplopia at 1.5˚ and 5.0˚ (i.e. category 2) is expected to have a large suppression scotoma in extent, but with a weak intensity, whereas a subject who fuses at 5.0˚ but suppresses at 1.5˚ (i.e. category 3) would have a smaller scotoma in extent but deeper in intensity. The second subject is more likely to exhibit higher binocular imbalance than the first. Nevertheless, this hypothesis was not confirmed, we believe that due to the size of the stimuli used in the VR test (4.4˚).

In a second analysis of the VR test validity, we found significant correlations between the binocular imbalance results and both stereoacuity and LogMAR interocular acuity differences when considering the whole dataset (Table 2). Inter-observer discriminatory power provides a third line of evidence for the validity of the binocular imbalance test (discussed below).

The data collected for this study are representative of the problems faced in clinics where amblyopic patients are seen. Most patients have received prior treatment for amblyopia, focused on recovering visual acuity, using occlusion (59 volunteers; 11 out of 24 subjects in the anisometric group; 25 out of 30 in the mixed group; and 23 out of 25 in the strabismus group). Mean interocular visual acuity differences are relatively low thanks to those previous treatments, but the visual problem persists as stereoacuity and the imbalance factor values show (Table 3, Fig 3). According to the standard acuity definition of amblyopia, only 27 participants would be considered to be amblyopic. However, when the other variables of the triplet are considered, the presence of a visual loss is clearly manifested.

The three variables of the triplet have different behaviors depending on group classification. The 3 variables are correlated for the whole dataset and, in several cases, when considering the clinic subgroups of patients (Table 2). Nevertheless, their mean values differ significantly, depending on group classification (Table 3). These results confirm and extend previous attempts to compare clinical subgroups using binocular imbalance, acuity differences and stereoacuity [19].

Anisometropic patients show clearly worse stereoacuity and larger acuity differences than the control group (Table 3). As previously shown, most purely anisometropic patients retain some stereopsis [8] (only 2 out of 24 participants had no measurable stereoacuity). Anisometropic amblyopes have stereopsis at low, but not high, spatial frequencies, suggesting that while their stereoacuity is not as acute as normal, it is nevertheless functional [12]. Anisometropic patients show low imbalance factors (mean value 1.5, meaning that the amblyopic eye needs 1.5 times more contrast than the fellow eye to achieve balanced vision), only slightly higher than the control group (Table 3). The imbalance factor fails to differentiate anisometropic patients from controls (means pair comparison, Table 4). Accordingly, our logistic model shows poor results when imbalance factor is used to detect anisometropic patients (ROC Area under the curve 0.59), compared with stereoacuity or acuity difference models (Table 5). The regional extent and depth of suppression in patients with anisometropia did not differ from controls in previous studies when they were assessed at low spatial frequencies, as in the current study (0.68 cpd) [23,51]. However, in amblyopic patients, binocular imbalance increases with spatial frequency and the factor could be as much as 2–8 at 2.72 cpd [23,24]. In the future, improved resolution in VR headsets may allow testing at higher spatial frequencies, and this might help to differentiate anisometropic patients.

On the other hand, the imbalance factor provides important information for characterizing strabismic and mixed groups [25] (mean imbalance factor amounts to 4.4 for mixed and 3.5 for strabismic, whereas it is only 1.5 for anisometropia, Table 3). Suppression in strabismus may be different from suppression in anisometropia, active in the former, to avoid diplopia, and more passive in the latter, because of visual acuity loss in the amblyopic eye [1]. Stereopsis is also more impacted in strabismic than in anisometropic amblyopia [1,8], as our data

corroborates: mean value of log10 stereoacuity in anisometropia is 2.1, whereas mixed and strabismic groups show 2.7 and 2.9 respectively (Table 3).

The logistic model using stereoacuity or imbalance factor as predictors shows strong differentiation among subgroups (Table 5). However, a high proportion of strabismic (17 out of 25) and mixed (17 out of 30) participants fall into the category of stereoblind. The imbalance factor variable would allow tracking a patient's evolution on the path to recovering stereovision in much more subtle detail, especially when using a dichoptic training approach [1,14,52]. Thus, stereoacuity is a good red flag for detecting the presence of an amblyogenic factor, but it is not the best variable for quantifying the severity of the problem. On the other hand, the imbalance factor is a good detection variable according to the logistic model, and solves the clinic problem of quantifying the severity of the problem.

Interestingly, whereas mean differences are not significant for any variable of the triplet between strabismic and mixed groups (Table 4), all show significant differences between accommodative and non-accommodative subgroups. A patient whose deviation can be resolved at near and far just using the appropriate refraction is likely to show a much better baseline triplet: low dominance and gross stereoacuity (only 1 out of 16 accommodative patients is stereoblind). This seems reasonable, as both images are reasonably well correlated, and should fall within Panum's area [53]. On the other hand, if the refractive correction does not completely resolve the deviation, there is a high chance of being stereoblind (33 out of 39 in our sample are stereoblind) and having a high imbalance factor. Accommodative patients do not manifest strabismus and therefore are receiving spatially concordant binocular visual inputs for most of their waking hours. Consequently, they have a better prognosis than non-accommodative strabismus patients [54]. Although different in etiology, both anisometropia and accommodative groups show a similar baseline triplet. In fact, accommodative and anisometropia groups show only mild mean differences in binocular imbalance (Table 4).

## Conclusions

The binocular imbalance test, using a Virtual Reality device, is easy for patients to understand, fast, repeatable and valid. VR is a promising technique in amblyopia treatment. Adjusting contrast to rebalance binocular vision within a VR headset opens the possibility of new treatments based on this technology.

Our results stress the importance of monitoring amblyopia in clinical practice not only taking into account visual acuity, but also stereoacuity and interocular imbalance. Patching and visual therapy outcomes should be tracked using this triplet. Amblyopia is not only a monocular disorder, but also a binocular problem. Stereoacuity on its own is not sufficient to quantify the effects of treatment, because of the high percentage of patients with no measurable initial stereoacuity. Measuring binocular imbalance may help to fill that gap.

Each amblyogenic risk factor (anisometropia, strabismus or mixed) has a different representation in the triplet space. Moreover, the results emphasize the importance of determining the nature of the deviation: accommodative strabismus has a better prognosis than non-accommodative strabismus.

## Supporting information

**S1 Table. Raw data.**
(CSV)

**S1 Movie. Binocular imbalance test in VR.**
(MP4)

## Acknowledgments

We would like to thank Iñaki Basterra Barrenetxea for his contribution in the recruitment of patients.

## Author Contributions

**Conceptualization:** Santiago Martín, Juan A. Portela, Jian Ding.

**Data curation:** Juan A. Portela.

**Formal analysis:** Oliver Ibarrondo.

**Investigation:** Santiago Martín, Jian Ding.

**Methodology:** Santiago Martín, Juan A. Portela.

**Resources:** Juan A. Portela.

**Software:** Santiago Martín.

**Supervision:** Juan A. Portela, Jian Ding, Dennis M. Levi.

**Validation:** Juan A. Portela, Dennis M. Levi.

**Writing – original draft:** Santiago Martín.

**Writing – review & editing:** Juan A. Portela, Jian Ding, Dennis M. Levi.

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
