## [Decision Letter · Decision Letter 0]

26 Jun 2020

PONE-D-20-15817

Evaluation of a Virtual Reality implementation of a binocular imbalance test in a large clinic population

PLOS ONE

Dear Dr. Martin Gonzalez,

Thank you for submitting your manuscript to PLOS ONE. After careful consideration, we feel that it has merit but does not fully meet PLOS ONE’s publication criteria as it currently stands. Therefore, we invite you to submit a revised version of the manuscript that addresses the points raised during the review process.

The authors should provide more details with illustrated figures to better demonstrate how they used virtual reality interface.

We look forward to receiving your revised manuscript.

Kind regards,

Ahmed Awadein, MD, Ph.D, FRCS

Academic Editor

PLOS ONE

Journal Requirements:

"I have read the journal's policy and the authors of this manuscript have the following competing interests: SM and JP promoted, with the support of the University of Oviedo, the creation of the startup VisionaryTool. Both have assisted VisionaryTool, S.L. (www.visionarytool.com) to create a commercial version of the VR imbalance test described in this manuscript (University of Oviedo contract FUO-EM-19-099). VisionaryTool has not had any role (writing, analysis, or control over publication) in the production of the paper."

Reviewers' comments:

Reviewer's Responses to Questions

**Comments to the Author**

1. Is the manuscript technically sound, and do the data support the conclusions?

Reviewer #1: Yes

Reviewer #2: Yes

2. Has the statistical analysis been performed appropriately and rigorously? 

Reviewer #1: Yes

Reviewer #2: Yes

3. Have the authors made all data underlying the findings in their manuscript fully available?

Reviewer #1: Yes

Reviewer #2: Yes

4. Is the manuscript presented in an intelligible fashion and written in standard English?

Reviewer #1: Yes

Reviewer #2: Yes

5. Review Comments to the Author

Reviewer #1: The topic is interesting and a new revolution in the management of amblyopia. It is, however, tough and understood with a great effort, which the reader might not always with to spend. I suggest adding more figures of the materials used and steps done to further clarify the topic to the reader. I also suggest more detailed explanation of the principle of the test and virtual reality in a simplified way. In general, it is well written and high ranking but difficult for an average reader to understand readily. Try to simplify it and go step by step wit the reader, while using the aid of more figures.

Reviewer #2: Authors did a good work with the VR device. This trend may be the future of amblyopia therapy. Here are few comments.

Line 26: “The triplet”: I assume the authors mean by the triplet the three factors together as the word in mentioned again in line 31. It is not readily clear to readers what the triplet in line 26 means. Please clarify.

Lines 55-57: “The absence of stereoacuity…. Treatment”: This might be a bit of an overstatement and not fully supported by the cited papers. There is uncertainty in measuring stereoacuity in amblyopes, and there is uncertainty in following improvement in stereoacuity. However, this does not create major uncertainty in tracking the improvement in amblyopia in general.

Lines 74 – 75: “Black et al.. headset”: The references cited here are confusing. Reference 21 describes aligning both image by moving a line on the screen before stating the test. This particular study did not test or verify the technique used to compensate for misalignment. Authors may consider referring to another study that focus on this. Reference 30 at cited at the end of the sentence is a study about binocular influence on global motion processing and does not describe any use of VR headset.

Line 78: “VR combined”: Please explain what you mean by VR combined. The term does not appear to be common in literature or it may be a truncated sentence.

Lines 86: Please mention the reference (Kwon et al) in the regular fashion not in line with the text.

Lines 86: The first aim of the study is to “implement a test .. in a VR headset”. It is understandable to readers that this test was designed and published by Kwon et al in 2014 and the authors are using this test in their VR devise. I would have expected the authors to dedicate few lines of their introduction describing this test and why did they select this particular test to use in their device. I assumed that they meant reference number 26. This should be made easier for readers to find. This is the highlight of the introduction and I believe authors could have spent less time going over all the progress made by VR devices and focus more on describing the tests that that will use and aim to implement in a mainstream VR device.

Line 88 and 90: “Large population of clinical patients”: I would not consider 100 volunteers (of which 21 are controls) recruited in one optometry clinic a large population of clinical patients.

Line 98: Inclusion criteria should come before the exclusion.

Line 103: Authors do not state who were they recruiting as cases (not control). Were they recruiting any patient with amblyopia? Or any patient with amblyogenic factor even if there was no amblyopia present? Or any patient with a history of amblyopia as taken from previous files? Where is the cases selection criteria?

Line 110: “All 55 participants with strabismus had esotorpia”: This needs thorough explanation. In the age group described it is expected to have a good number of exotropia cases. Some may argue that you may even expect to see more exotropia than esotropia. The only was the authors ended up with 55 amblyopes with strabismus and the 55 had esotropia is that they were selecting esotropia and excluding exotropia and we don’t see that in their exclusion criteria. And if that was the case, why only choose esotropes? Moreover, the authors did not exclude patients with previous strabismus surgery. In the raw data table the only surgery mentioned is cataract surgery. Again, it is very unlikely in such group with strabismus in this age group to have no surgical history unless you are excluding surgical cases.

Lines 115 – 116: “Anisometropia was defined ... >=1 D”. Where did the authors get this definition? Reference 39 cited at the end of this sentence is an article describing the effect of amblyopia treatment on stereoacuity and although they used 1D of interocular difference to define anisometropia, yet, they did not investigate if this number was significant. A one diopter difference in a bilateral myope of 8 and 9 D is not like a 1 D difference in a unilateral hypertmetropa of 1D. Lumping up all the anisometropes together under 1 D may not be clinically relevant.

Lines 125 -126: The UCT test for strabismus is repeated here. It was described few lines above.

Line 217: Did the authors use prisms while doing the W4D test on strabismic patients? Authors corrected for misalignment in the VR test. In order for the W4D and the VR test to be comparable, both have to correct (or not correct) for misalignment.

Line 233: This categorization of the W4D result is interesting. If it is has been described before please add the reference. If it is devised by the authors please explain what made you come up with this grading. For example: why would someone with diplopia at both angles be classified as better than someone who can fuse at 5 degs but suppress at 1.5 degs. If this diplopia is manifest then definitely his binocularity and stereo should be graded less than a patient who can fuse for near.

Line 308: “but not only” Consider revising the sentence structure

Line 311: “To the extend” Consider revising the sentence structure.

Lines 488 -489: Year of publication of the article missing.

6. PLOS authors have the option to publish the peer review history of their article (what does this mean?). If published, this will include your full peer review and any attached files.

Reviewer #1: **Yes: **Rehab R Kassem

Reviewer #2: No

---

## [Author Response · Author response to Decision Letter 0]

23 Jul 2020

EDITOR

We have reviewed the style requirements, particularly regarding reference citations

Please include an updated Competing Interests statement in your cover letter

We have added the sentence "This does not alter our adherence to PLOS ONE policies on sharing data and materials” as requested in the cover letter.

The authors should provide more details with illustrated figures to better demonstrate how they used virtual reality interface.

We have included a video in the complementary materials section to explain how virtual reality interface works. We have also modified figure 1 to explain better the proposed test. 

Reviewer #1

The topic is interesting and a new revolution in the management of amblyopia. It is, however, tough and understood with a great effort, which the reader might not always with to spend. I suggest adding more figures of the materials used and steps done to further clarify the topic to the reader. I also suggest more detailed explanation of the principle of the test and virtual reality in a simplified way. In general, it is well written and high ranking but difficult for an average reader to understand readily. Try to simplify it and go step by step wit the reader, while using the aid of more figures.

We have included a video in the complementary materials section to explain how virtual reality interface works. We have also modified figure 1 to better explain the proposed test. We have added in the Introduction section a description of the test proposed by Kwon et al in 2015 and we have also explained why we selected this test specifically as benchmark for the VR test.

 

Reviewer #2: Authors did a good work with the VR device. This trend may be the future of amblyopia therapy. Here are few comments.

Line 26: “The triplet”: I assume the authors mean by the triplet the three factors together as the word in mentioned again in line 31. It is not readily clear to readers what the triplet in line 26 means. Please clarify.

We have changed line 26 to make it easier to understand. 

Lines 55-57: “The absence of stereoacuity…. Treatment”: This might be a bit of an overstatement and not fully supported by the cited papers. There is uncertainty in measuring stereoacuity in amblyopes, and there is uncertainty in following improvement in stereoacuity. However, this does not create major uncertainty in tracking the improvement in amblyopia in general.

The idea we want to stress is that if we wish to quantify the improvement in binocularity, the absence of an initial measure of stereoacuity and the poor test-retest reliability of many standard clinical tests, makes quantification difficult. We have changed the previous sentence in the paper to explain this idea more clearly, and we have added a reference to a recent paper by Webber et al. 2020 that deeps in the importance of dominance in amblyopia.

Lines 74 – 75: “Black et al.. headset”: The references cited here are confusing. Reference 21 describes aligning both image by moving a line on the screen before stating the test. This particular study did not test or verify the technique used to compensate for misalignment. Authors may consider referring to another study that focus on this. Reference 30 at cited at the end of the sentence is a study about binocular influence on global motion processing and does not describe any use of VR headset.

Reference [30] was included in this sentence because it explains the fundamentals of the dichoptic binocular imbalance test used in reference [21]. Nevertheless, it might be confusing as the reviewer points out, so it has been removed. In order to explain better the importance of compensating any misalignment we have included a new reference focused on this topic and modified the text accordingly [A].

[A] Recovery of stereopsis through perceptual learning in human adults with abnormal binocular vision. Jian Ding and Dennis M. Levi. PNAS | September 13, 2011 | vol. 108 | no. 37 | E733–E741

Line 78: “VR combined”: Please explain what you mean by VR combined. The term does not appear to be common in literature or it may be a truncated sentence.

We apologize, the term “combined” is now deleted, it was an erratum.

Lines 86: Please mention the reference (Kwon et al) in the regular fashion not in line with the text.

We have modified the style for this reference

Lines 86: The first aim of the study is to “implement a test .. in a VR headset”. It is understandable to readers that this test was designed and published by Kwon et al in 2014 and the authors are using this test in their VR devise. I would have expected the authors to dedicate few lines of their introduction describing this test and why did they select this particular test to use in their device. I assumed that they meant reference number 26. This should be made easier for readers to find. This is the highlight of the introduction and I believe authors could have spent less time going over all the progress made by VR devices and focus more on describing the tests that that will use and aim to implement in a mainstream VR device.

We want to stress here that, due to our mistake, the date of the reference was wrong, as we meant reference number [27] (according to the first manuscript version) and not [26]. We have added in the Introduction section a description of the test proposed by Kwon et al in 2015 and we have also explained why we selected this test specifically as benchmark for the VR test.

Line 88 and 90: “Large population of clinical patients”: I would not consider 100 volunteers (of which 21 are controls) recruited in one optometry clinic a large population of clinical patients.

We have leave out the term “large” from this sentence and the title of the paper.

Line 98: Inclusion criteria should come before the exclusion.

Exclusion criteria has been moved to the end of Participant’s section.

Line 103: Authors do not state who were they recruiting as cases (not control). Were they recruiting any patient with amblyopia? Or any patient with amblyogenic factor even if there was no amblyopia present? Or any patient with a history of amblyopia as taken from previous files? Where is the cases selection criteria?

Inclusion criteria refers to patients with amblyopia or with a history of amblyopia that had been treated successfully. We have clarified this point in the Participant’s section and in the Abstract.

Line 110: “All 55 participants with strabismus had esotropia”: This needs thorough explanation. In the age group described it is expected to have a good number of exotropia cases. Some may argue that you may even expect to see more exotropia than esotropia. The only was the authors ended up with 55 amblyopes with strabismus and the 55 had esotropia is that they were selecting esotropia and excluding exotropia and we don’t see that in their exclusion criteria. And if that was the case, why only choose esotropes?

Esotropia (ET) is the most common form of strabismus, accounting between half to two thirds of all misaligned eyes [A]. Exotropia (XT) has low prevalence: when present, it takes the form of intermittent exotropia in most of the cases, while constant exotropia is much less common [B]. Binocular inhibition is low in intermittent XT [C]; and the deviation grade in near is lower than in far distances [D]. In fact, fine stereoacuity at near distance is a common clinical finding in intermittent XT. According to our clinical experience, amblyopia is uncommon in intermittent XT.

This explains why our strabismus sample only had ET, taking into account that we have included only patients with amblyopia or with a history of amblyopia.

We have clarified this point in the Participant’s section. 

[A] Pai A, Mitchell P. Prevalence of amblyopia and strabismus. Ophthalmology. 2010;117(10):2043–2044.

[B] Govindan M, Mohney BG, Diehl NN, Burke JP. Incidence and types of childhood exotropia: a population-based study. Ophthalmology. 2005;112(1):104-108. 

[C] Ahn SJ, Yang HK, Hwang JM. Binocular visual acuity in intermittent exotropia: role of accommodative convergence. Am J Ophthalmol. 2012;154(6):981.

[D] Mohney BG, Holmes JM. An office-based scale for assessing control in intermittent exotropia. Strabismus. 2006;14(3):147-150. doi:10.1080/09273970600894716

Line 110: Moreover, the authors did not exclude patients with previous strabismus surgery. In the raw data table the only surgery mentioned is cataract surgery. Again, it is very unlikely in such group with strabismus in this age group to have no surgical history unless you are excluding surgical cases.

Reviewer is right. The clinic does not practice strabismus surgery, but data was available in the anamnesis form. Six patients had previous strabismus surgery. This information is now included in the complementary material and cited in the text.

Lines 115 – 116: “Anisometropia was defined ... >=1 D”. Where did the authors get this definition? Reference 39 cited at the end of this sentence is an article describing the effect of amblyopia treatment on stereoacuity and although they used 1D of interocular difference to define anisometropia, yet, they did not investigate if this number was significant. A one diopter difference in a bilateral myope of 8 and 9 D is not like a 1 D difference in a unilateral hypertmetropa of 1D. Lumping up all the anisometropes together under 1 D may not be clinically relevant.

While the reviewer is correct, every published RCT (including the PEDIG studies) had used a fixed dioptric definition, generally >=1. We have adopted this definition so that our results can be readily compared with previous work.

Some references with the same criteria are:

[A] Wong TY, Foster PJ, Hee J, et al. Prevalence and risk factors for refractive errors in adult Chinese in Singapore. Invest Ophthalmol Vis Sci. 2000;41(9):2486-2494.

[B] Dobson V, Harvey EM, Miller JM, Clifford-Donaldson CE. Anisometropia prevalence in a highly astigmatic school-aged population. Optom Vis Sci. 2008;85(7):512-519. doi:10.1097/OPX.0b013e31817c930b

[C] Deng L, Gwiazda JE. Anisometropia in children from infancy to 15 years. Invest Ophthalmol Vis Sci. 2012;53(7):3782-3787. Published 2012 Jun 20. doi:10.1167/iovs.11-8727

[D] Afsari S, Rose KA, Gole GA, et al. Prevalence of anisometropia and its association with refractive error and amblyopia in preschool children. Br J Ophthalmol. 2013;97(9):1095-1099. doi:10.1136/bjophthalmol-2012-302637

[E] Lee C-W, Fang S-Y, Tsai D-C, Huang N, Hsu C-C, Chen S-Y, et al. (2017) Prevalence and association of refractive anisometropia with near work habits among young schoolchildren: The evidence from a population-based study. PLoS ONE 12(3): e0173519. https://doi.org/10.1371/journal.pone.0173519

Lines 125 -126: The UCT test for strabismus is repeated here. It was described few lines above.

We have shortened the sentence to avoid duplication.

Line 217: Did the authors use prisms while doing the W4D test on strabismic patients? Authors corrected for misalignment in the VR test. In order for the W4D and the VR test to be comparable, both have to correct (or not correct) for misalignment.

We do not state that W4D test and VR test measure the same problem. We just analyze the agreement between W4D test and the three variables used in the study (acuity, stereoacuity and binocular imbalance) proposing an original W4D scale. We want to analyze the coherence and differences between these different measurements. We find relevant W4D because it is widely used in daily clinic practice. 

W4D test was carried out without prisms in strabismic patients because we want to assess not only if there is suppression or if the patient can fuse, but also if there is diplopia under dissociated stimuli (the red/green dots used in W4D test are highly dissociative).

We should have stated in the exclusion criteria, the presence of diplopia in daily life conditions (now we have included this). If diplopia shows up in W4D test and not in daily life conditions, it means that there is suppression in daily life conditions, but the suppression scotoma is weak or small in extent. This idea is at the root of the scale propose for W4D test.

The VR test uses a virtual prism to avoid diplopia because we want to measure the degree of binocular imbalance. Surprisingly, even if we correct the deviation angle and we consider amblyopia treatment successful, we found an anomalous binocular imbalance.

As a result of this discussion we have modified the exclusion criteria section and made a deeper explanation of how W4D test was carried out.

Line 233: This categorization of the W4D result is interesting. If it has been described before please add the reference. If it is devised by the authors please explain what made you come up with this grading. For example: why would someone with diplopia at both angles be classified as better than someone who can fuse at 5 degs but suppress at 1.5 degs. If this diplopia is manifest then definitely his binocularity and stereo should be graded less than a patient who can fuse for near.

As explained before, if diplopia shows up in W4D test and not in daily life conditions, it means that there is suppression, but the suppression scotoma is weak. A subject could have diplopia at 5 degrees and suppression at 1.5 degrees. Or the patient could fuse at 5 degrees and suppress at 1.5 degrees. But will never fuse or exhibit diplopia at 1.5 and suppress at 5 degrees. Diplopia appears only due to the highly dissociative stimuli used in W4D test.

This idea is at the root of the scale propose for W4D test. Considering diplopia and performing the test at different angles, we have a broader scale than the one proposed by Webber et al. 

A subject with diplopia at both angles has a large suppression scotoma in extent, but with a weak intensity. A subject who fuses at 5 degrees but suppresses at 1.5 degrees has a smaller scotoma in extent but deeper in intensity. The second subject is more likely to exhibit higher binocular imbalance than the first. Nevertheless, with the data obtained in this study this hypothesis has not been confirmed. One possible reason is the size of the stimuli used in the VR test. As explained in the discussion chapter, we would like to use a smaller stimulus to measure binocular imbalance at higher frequencies. Binocular imbalance is more evident at high frequencies and the correlation with W4D test proposed scale is more likely to happen if the stimuli is smaller.

As a result of this discussion we have added in both the results and discussion sections deeper considerations about the proposed W4D test score.

Line 308: “but not only” Consider revising the sentence structure

Revised

Line 311: “To the extend” Consider revising the sentence structure.

Revised

Lines 488 -489: Year of publication of the article missing.

Corrected

---

## [Editor Report · Decision Letter 1]

10 Aug 2020

Evaluation of a Virtual Reality implementation of a binocular imbalance test

PONE-D-20-15817R1

Dear Dr. Martin Gonzalez,

We’re pleased to inform you that your manuscript has been judged scientifically suitable for publication and will be formally accepted for publication once it meets all outstanding technical requirements.

Kind regards,

Ahmed Awadein, MD, Ph.D, FRCS

Academic Editor

PLOS ONE
---

## [Editor Report · Acceptance letter]

13 Aug 2020

PONE-D-20-15817R1 

Evaluation of a Virtual Reality implementation of a binocular imbalance test 

Dear Dr. Martín:

I'm pleased to inform you that your manuscript has been deemed suitable for publication in PLOS ONE. Congratulations! Your manuscript is now with our production department. 

Kind regards, 

on behalf of

Dr. Ahmed Awadein 

Academic Editor

PLOS ONE